# CNN-Based Approaches with Different Tumor Bounding Options for Lymph Node Status Prediction in Breast DCE-MRI

**DOI:** 10.3390/cancers14194574

**Published:** 2022-09-21

**Authors:** Domiziana Santucci, Eliodoro Faiella, Michela Gravina, Ermanno Cordelli, Carlo de Felice, Bruno Beomonte Zobel, Giulio Iannello, Carlo Sansone, Paolo Soda

**Affiliations:** 1Unit of Computer Systems and Bioinformatics, Department of Engineering, University of Rome “Campus Bio-medico”, Via Alvaro del Portillo, 21, 00128 Rome, Italy; 2Department of Radiology, Sant’Anna Hospital, Via Ravona, 22042 Como, Italy; 3Department of Electrical Engineering and Information Technology, University of Naples Federico II, 80131 Naples, Italy; 4Department of Radiology, University of Rome “Sapienza”, Viale del Policlinico, 155, 00161 Rome, Italy; 5Department of Radiology, University of Rome “Campus Bio-medico”, Via Alvaro del Portillo, 21, 00128 Rome, Italy; 6Department of Radiation Sciences, Radiation Physics, Biomedical Engineering, Umeå University, Universitetstorget, 490187 Umeå, Sweden

**Keywords:** breast cancer (BC), axillary lymph nodes status (ALNS), deep learning (DL), convolutional neural network (CNN), bounding box

## Abstract

**Simple Summary:**

Breast cancer represents the most frequent cancer in women in the world. The state of the axillary lymph node is considered an independent prognostic factor and is currently evaluated only with invasive methods. Deep learning approaches, especially the ones based on convolutional neural networks, offer a valid, non-invasive alternative, allowing extraction of large amounts of the quantitative data that are used to build predictive models. The aim of our work is to evaluate the influence of the peritumoral parenchyma through different bounding box techniques on the prediction of the axillary lymph node in breast cancer patients using a deep learning artificial intelligence approach.

**Abstract:**

Background: The axillary lymph node status (ALNS) is one of the most important prognostic factors in breast cancer (BC) patients, and it is currently evaluated by invasive procedures. Dynamic contrast-enhanced magnetic resonance imaging (DCE-MRI), highlights the physiological and morphological characteristics of primary tumor tissue. Deep learning approaches (DL), such as convolutional neural networks (CNNs), are able to autonomously learn the set of features directly from images for a specific task. Materials and Methods: A total of 155 malignant BC lesions evaluated via DCE-MRI were included in the study. For each patient’s clinical data, the tumor histological and MRI characteristics and axillary lymph node status (ALNS) were assessed. LNS was considered to be the final label and dichotomized (LN+ (27 patients) vs. LN− (128 patients)). Based on the concept that peritumoral tissue contains valuable information about tumor aggressiveness, in this work, we analyze the contributions of six different tumor bounding options to predict the LNS using a CNN. These bounding boxes include a single fixed-size box (SFB), a single variable-size box (SVB), a single isotropic-size box (SIB), a single lesion variable-size box (SLVB), a single lesion isotropic-size box (SLIB), and a two-dimensional slice (2DS) option. According to the characteristics of the volumes considered as inputs, three different CNNs were investigated: the SFB-NET (for the SFB), the VB-NET (for the SVB, SIB, SLVB, and SLIB), and the 2DS-NET (for the 2DS). All the experiments were run in 10-fold cross-validation. The performance of each CNN was evaluated in terms of accuracy, sensitivity, specificity, the area under the ROC curve (AUC), and Cohen’s kappa coefficient (K). Results: The best accuracy and AUC are obtained by the 2DS-NET (78.63% and 77.86%, respectively). The 2DS-NET also showed the highest specificity, whilst the highest sensibility was attained by the VB-NET based on the SVB and SIB as bounding options. Conclusion: We have demonstrated that a selective inclusion of the DCE-MRI’s peritumoral tissue increases accuracy in the lymph node status prediction in BC patients using CNNs as a DL approach.

## 1. Introduction

Breast cancer (BC) is the most frequent cancer in women worldwide [1]. The axillary lymph node status (ALNS) is considered one of the most influential and independent prognostic factors [2]. For accurate lymph node staging, invasive procedures from sentinel node biopsy to axillary dissection are needed. The positive predictive value (PPV) of lymph node cytology and biopsy is about 99–100%, however, a lower negative predictive value, about 65–83%, does not permit the avoidance of axillary surgery in negative cases [3]. 

Although some imaging techniques are used for the detection of nodal metastasis, such as mammography, ultrasonography (US), and computed tomography (CT), currently, there has not been an agreement on the imaging criteria for identifying metastatic nodes. Size and shape are generally used for LNS prediction (dense nodes of increased attenuation and spiculated contour on mammography, a short-axis dimension greater than 10 mm on CT, a long-axis dimension greater than 5 mm, a round shape, and a long axis greater than 5 mm on US [4,5,6,7,8]

Moreover, among all the employed imaging methods, magnetic resonance imaging (MRI) performed best for primary tumor analysis, providing qualitative and quantitative information [9,10,11], especially of the dynamic contrast-enhanced sequences (DCE), which, thanks to the high contrast resolution, allow the best depiction of the tumor morphology, size and perfusional behavior, distinction between benign and malignant lesions, prediction of biological aggressiveness, and prognostic evaluation [9,12]. 

In recent years, imaging-based machine learning (ML) techniques have been employed in many oncological fields with promising results to support medical decisions. In this area, ML contemplates two different approaches: The first, usually referred to as radiomics, is based on handcrafted features computed from the images feeding shallow learners, such as support vector machines, decision trees, etc. The second is based on deep neural networks and is known as deep learning (DL). Different from the first approach, DL automatically learns discriminative features directly from images without being limited to the use of predefined features or the developers’ experience.

Among the DL-based solutions used to analyze visual imagery, the use of convolutional neural networks (CNNs) is the most applied. A CNN consists of a set of convolutional layers capable of learning a compact hierarchical representation of the input that is well suited to the specific task to be solved. 

In BC, radiomics has been studied considerably for the characterization of primary cancer, while there are still relatively few works that evaluate its application to the relationship between tumor imaging characteristics and LNS [13,14,15]. In these works, in particular, traditional radiomics systems were mainly investigated, while DL applications are limited to a few studies and ultrasound cases [16,17,18]. Furthermore, the majority of these studies focused on analyzing the imaging features of the primary tumor exclusively, ignoring the tissue adjacent to the tumor. Nevertheless, evidence has shown that the peritumoral region contains valuable information on the potential tumor aggressiveness and, also, lymphatic spread, in particular in cases of multifocal and multicentric tumors [19,20,21]. In our previous works [13,19], we analyzed the impact of the radiomics of the primary tumor and the peritumoral edema on the LNS prediction. This time, based on the previously cited papers demonstrating the importance of the peritumoral parenchyma, we assume that, close to the tumor lesions, there might be some data not visible to the human eye that influences the metastatic lymph node spread.

Then, in this work, we investigate the role of the lesions neighboring parenchyma features through CNNs by experimentally analyzing different tumor bounding (TB) options. The novelty of the technical contribution can be summarized as follows: We propose and implement different tumor bounding options to systematically assess the contribution gathered by the healthy tissue in the prediction of the axillary lymph node status;We propose three different CNNs whose architectures vary according to the characteristics of the volumes considered as inputs;We evaluate the impact of the 3D and 2D features extracted from input volumes and the influence of healthy tissue in cases of multifocal and multicentric tumors.

## 2. Study Population 

All breast MRI exams were performed for preoperative evaluation at the Central Radiology Department of Policlinico Umberto I between January 2017 and January 2020 and retrospectively reviewed. A written informed consensus was obtained before the execution of a contrast-MRI for all examinations. 

All patients met the following inclusion criteria: three Tesla magnetic field MRI examinations, post-contrast sequences, mass-like tumors, histopathological confirmation of invasive breast cancer, a complete histological analysis, and definitive lymph-node status of the ipsilateral axilla. In cases of BC bilateral lesions, the two lesions were evaluated separately since the two breasts can be considered a single part. Patients with an incomplete MRI examination or damaged images and patients without a complete histopathological analysis were excluded.

The patients were excluded if they had breast implants or expanders, were in follow-up neo- or adjuvant chemotherapy, or the MRI images were not of excellent diagnostic quality. 

A total of 153 patients (average age 55 years; range 30–85) met the inclusion criteria. In two patients who had bilateral breast cancer, the two breasts were considered as a single, independent part. Therefore, a total of 155 malignant breast cancer lesions were included in total. 

The LNS status was assessed as positive if at least one lymph node involved by metastasis was present in the definitive histopathological axillary cable sample (LN+); the LNS was considered negative if all axillary lymph nodes were safe (LN−).

### 2.1. MRI Examination 

All MRI examinations were performed using a 3T magnet (Discovery 750; GE Healthcare, Milwaukee, WI, USA). The sequence used for the analysis was the dynamic Contrast-enhanced T1-weighted 3D sequence using fat suppression with a TR and a TE of 6.6 ms and 4.3 ms, respectively, with an ip angle of 10°, a matrix of 512 × 256, NEX 1, and a slice thickness of 2.4 mm. An amount of 0.2 mmol/kg of Gadobenate-dimeglumine (Multihance^®^; Bracco Imaging, Milan, Italy) was used as the contrast agent, injected through a 20G intravenous cannula at a rate of 2 mL/s plus 15 mL of saline solution at the same speed. For each acquisition, the relative subtracted images were automatically generated and used for tumor analysis.

The images were analyzed by two radiologists with 10 and 3 years of experience, respectively. The tumors were described as unifocal when only one lesion was present; multifocal when more than one tumor lesion was present in the same breast quadrant/region; and multicentric when multiple tumor lesions were present in different breast quadrants/regions. For each lesion, the target dimensions, margins (regular, irregular, lobulated, or spiculated), and intensity signal timing curve (I, II, or III, based on wash-in and wash-out) were reported.

### 2.2. Clinical Data 

The patients’ clinical data were collected, and, according to these data, the population was split into subgroups: age, familiarity (considered positive if at least one familiar member was affected by breast cancer at any age), hormone therapy (considered positive if the patient performed at least 3 continuous months of hormone therapy including any kind contraceptive, replacement, or therapeutic therapy), and menopausal status.

### 2.3. Histological Data 

The samples were obtained by a core biopsy or surgery and analyzed by an anatomic pathologist with more than 15 years of experience. The tumor histotype classification followed the WHO classification [22]. The tumor histological grade was assigned in accordance with the NGS, and a score from one to three was given for these tumor characteristics: tubular formation, nuclear pleomorphism, and the number of mitoses. Furthermore, the estrogen receptor (ER), progesterone receptor (PgR), human epidermal growth factor receptor (HER2), and the proliferation index Ki67 were assessed for immunohistochemical analysis. A cut-off of 10% was used to consider the ER and PgR expression as positive; while HER2 was considered positive when >+2, and ki67 was considered positive when >14%.

Moreover, other histological data were collected: histotype (including ductal carcinoma (IDC) and invasive lobular carcinoma (ILC)), grading (divided into G1, G2, or G3), and tumor class, which includes the hormone receptor status and the proliferation index percentage (Luminal A: ER+, HER2− and low ki67; Luminal B: ER+, HER2 −/+ and high ki67; HER2 overexpressed; Triple Negative (TN): ER−, PgR−, HER2−). 

### 2.4. Axillary Lymph Node Status

The axillary lymph node status was considered as the final output. The LNS was assessed after an invasive breast cancer diagnosis using definitive surgical characterization (sentinel node dissection, sampling dissection, or total lymphadenectomy, based on surgeon decision but curative in all cases) [1,2]. The LNS was simply classified as positive, if there was at least a sentinel LN involved, or negative if there was no positive lymph node. On this basis, the dataset accounts for 27 positive and 128 negative patients, which are referred to as LN+ and LN− in the following. Table 1 reports the details of the dataset used in this paper. 

### 2.5. Pre-Processing

The patients’ clinical data and the frequencies of the tumor histological and MRI characteristics were reported. The Wilcoxon test was performed to compare these data between patients with and without LN involvement, setting the statistical significance at *p* < 0.05. The statistical analysis was performed using MatLab v. 2020b [23].

### 2.6. Segmentation 

The images were anonymized and uploaded on a dedicated open-source software (3D Slicer, version 4.8), ([24], November 2012). An identification number (ID) was assigned to each patient. Bilateral tumors were considered with two different IDs. For each case, the subtracted post-contrast T1w-MRI was selected. The second phase (60–120 s) was selected for ROI segmentation, due to its higher contrast resolution. Then, a label map was created. The lesions were manually drawn through manual and assisted thresholding segmentation techniques on the axial projection (Figure 1). When present, necrosis was avoided by segmentation. For multifocal or multicentric tumors, all lesions, even the smallest, were segmented.

## 3. Deep Learning Analysis 

An assessment of axillary lymph nodes reflects inherent primary tumor features, whose examination enables the discovery of non-invasive substitutes for the sentinel node biopsy currently being utilized. Most literature proposals have focused on the DCE sequence-extracted hand-crafted characteristics of breast lesions and performed the classification with ML techniques. The absence of a well-defined, efficient collection of features in the area of BC has, however, prompted researchers to investigate large and heterogeneous characteristics, implementing a feature selection step to pick the most discriminative ones. 

The use of CNNs for the ALN metastasis prediction in this study is made possible by their capacity to automatically learn the set of features that best suits the problem at hand, thereby eliminating the need for the feature extraction stage common to ML techniques and resolving the challenge of identifying the most discriminating set of primary tumor characteristics. Furthermore, we investigate several tumor boundary alternatives that vary depending on the quantity of healthy tissue to include in order to assess how the peritumoral area influences the performance of the involved networks.

The implemented analysis consists of three main steps: the imaging data definition, used to prepare data belonging to the DCE sequences of different patients; the volume extraction and bounding options, which describe the tumor bounding options; and the architecture of the CNNs for the ALN assessment, which introduces the involved CNNs that differ according to the characteristics of the volumes considered as input.

### 3.1. Imaging Data Definition

The DCE acquisition consists of MRI images (3D volumes) taken before (pre-contrast) and after (post-contrast) the contrast agent (CA) injection, resulting in 4D data, with three spatial (x,y,z) dimensions and one temporal (t). Although MRI exams are acquired with the same instruments, patients may present a different number of acquired volumes, resulting in a need for the selection of a subset of them. By denoting the number of acquired volumes for a patient p with N_p_, with t_i_ the i_th_ acquisition, where t_0_ and $t_{N_{p}}$−1 are the pre-contrast and the last one, respectively, the subtractive series is obtained by considering t_i_ − t_0_ with i form 1 to N_p_ − 1. To equalize the number of acquisitions for all patients, four specific subtractive volumes are selected: the first (t_1_), second (t_2_), and last ($t_{N_{p}}$−1) volumes and the median index (t_m_) volume between the third and the second-to-last volume. In this way, information about the wash-in and wash-out of the CA due is preserved. 

### 3.2. Volume Extraction and Bounding Options

Based on the hypothesis that peritumoral tissue may contain valuable information on tumor aggressiveness, and thus affects the ALN metastasis spread [19,20,21], we evaluate how the amount of the included non-tumor tissue impacts the ALN involvement, considering that patients differ in the size and number of lesions. In particular, a total of six different tumor bounding options are proposed and analyzed. They differ according to their included quantities of healthy tissue, which allows us to evaluate the contribution gathered from the area surrounding the tumor. Figure 2 shows the differences in the axial projections of the proposed bounding options.

In the single fixed-size box (SFB), a fixed-size 3D bounding box is centered in the tumor, completely encompassing the whole tumor region (or of all lesions, in cases of multifocal or multicentric tumors), and is used to crop each subject. The bounding box is always the same; it is applied to all four subtractive acquisitions (t_1_, t_2_, t_m_, $t_{N_{p}}$−1) and is patient-independent. The amount of non-tumor tissue directly depends on the tumor lesions’ dimensions: in the case of a single and small lesion, the extracted volume contains a high portion of healthy tissue compared to a damaged one and vice versa.

In the single variable-size box (SVB), the smallest 3D cubical bounding box is used to crop each subject in the four subtractive selected acquisitions. In contrast to the SFB, the cubical box in the SVB option is patient-dependent and aims to limit the amount of included non-tumor tissue. As a consequence, the amount of non-tumor tissue depends on the shape of each patient’s tumor region and he difference between the largest and smallest dimensions. Nevertheless, in cases of multifocal and multicentric tumors, the parenchyma between lesions is included in the extracted 4D volume.

As aforementioned, the DCE sequence of a patient is 4D data, in which each voxel is associated with information regarding its measurement in millimeters. In more detail, the attribute of pixel spacing is the physical distance between the centers of each two-dimensional pixel, specified by two numeric values representing the row and vertical spacing, while the spacing between slices measures the spacing between slices along the normal to the first image. The above attributes represent the DCE image resolution. Since the DCE sequences belonging to different subjects may have different resolutions, in the single isotropic-size box (SIB) option all the volumes are firstly re-sampled to obtain MRI images with isotropic voxels, and then the tumor bounding option SVB is applied, as described above.

In the single lesion variable-size box (SLVB), the SVB option was applied to each tumor lesion of the considered patient. A box for each tumor lesion is extracted, and, therefore, the tissue between lesions is excluded in the prediction of the ALN status with the aim of analyzing how much the parenchyma between lesions in multifocal/multicentric tumors impacts the ALN assessment.

In the single lesion isotropic-size box (SLIB), the tumor bounding box involves extracting a different box for each of the involved patient’s lesions that considers a DCE sequence with 3D volumes re-sampled to obtain isotropic voxels. The SLVB procedure is applied after having resized each acquisition.

In the two-dimensional slice (2DS) option, we propose applying the SVB procedure and then cutting the sequence of the 3D cropped volumes along the projection with the highest spatial resolution, which results in a series of two-dimensional slices with four temporal instants. For each patient, this process generates a set of 3D slices, with two spatial dimensions and one temporal, which represents the same section of tissue seen at four different time points (t_1_,t_2_, t_m_, and $t_{N_{p}}$−1). It is worth noting that only the slices containing lesions are taken into account: this is possible since the ALN status assessment requires lesion segmentation/detection.

Table 2 summarizes the six different bounding options proposed in this paper.

### 3.3. CNN Architecture for ALN Assessment

Based on these different bounding box options, we investigate different CNNs, whose architectures were developed for turning to the characteristics of the volumes considered as inputs, for their ALN status predictions. Hence, we designed three CNNs: the SFB-NET, used for the SFB option; the VB-NET, considered when the size of the bounding box varies according to each tumor lesion, that is, for the SVB, SIB, SLVB, and SLIB; and the 2DS-NET, introduced for the 2DS bounding option. Each CNN receives four-channel volumes as input, which represents the considered acquisitions of the DCE sequence. The proposed networks consist of different reduction layers and two fully connected layers. The output of each CNN consists of a two-element vector, which, after the application of the softmax function, is interpreted as the probability that the network associates with each class, namely the positive (LN+) and negative (LN−) ones.

The SFB-NET is a 3D CNN with three reduction layers, whose architecture is shown in Figure 3a. Each reduction block consists of a convolutional layer with 5 × 5 × 5 kernels (the number of kernels depends on the output channels) and a stride set to four.

The architecture of the VB-NET is shown in Figure 3b. It is a 3D CNN, consisting of five reduction blocks. The 4D input volume represents the smallest cubical box surrounding the tumor region or each lesion considered at four different time instants. Therefore, the sequence of the five convolutional layers with 4 × 4 × 4 kernels aims to implement a gradual dimensionality reduction. The stride is set to two, and the padding is one, except for the last convolutional layer.

The 2DS-NET is reported in Figure 3c. It presents the same characteristics as described for the VB-NET and is implemented by replacing the 3D convolutions with the standard 2D ones. In other words, the 2DS-NET is a 2D CNN since the input is a 3D volume that represents a bidimensional slice with four temporal instants (channels). Each convolutional layer consists of 4 × 4 kernels, and the stride is set to two. The padding is one, except for the last convolutional layer.

### 3.4. Experimental Setup

The proposed tumor bounding box options result in volumes with different characteristics. Indeed, in the SFB case, a 160 × 160 × 80 box centered in the lesion center is used, generating a 160 × 160 × 80 × 4 volume for each patient, which represents the input for the SFB-NET. The size of the box is chosen in such a way that, for each patient, the tumor is completely enclosed. Since the SVB, SIB, SLVB, and SLIB options consider a bounding box size that varies according to each patient’s tumor region, a resize stage with the bilinear interpolation method provides a standardized size of 64 × 64 × 64 × 4, which then fed the VB-NET. The resize stage is also needed for the 2DS option. The extracted four-channel images are resized to a common size of 64 × 64 × 4, before consideration as inputs for the 2DS-NET.

As aforementioned, the SLVB, SLIB, and 2DS bounding options generate a set of volumes or 3D slices (four-channel images) for each patient. However, the aim is to provide a unique label for each subject that represents the potential risk of having axillary lymph node metastasis. As a consequence, the predicted classes of all volumes (or 3D slices) belonging to the same patient need to be combined. From all of the combining strategies, majority voting (MV) is used, in which the label for each subject is the most common predicted class over all of its volumes (or 3D slices).

We select the implemented CNN architectures by choosing the best configurations obtained by varying the size of the convolutional kernels from three to seven and the number of reduction layers from three to five. Moreover, we also take into account the architecture proposed by Nguyen et al. [25], which is the only architecture that aims to predict ALN metastasis using the primary tumor DCE-MRI features with a DL approach. In particular, we consider the solution proposed in [25] as a starting point for providing different changes.

The performance is evaluated in terms of accuracy (ACC), sensitivity (SENS), specificity (SPE), area under the ROC curve (AUC), and Cohen’s kappa coefficient (K). All experiments were run by applying a patient-wise 10-fold cross-validation (CV) to better assess the generalization ability of each approach, avoiding the use of volumes belonging to the same subject both in the training and evaluation steps. We split the set of patients into 10 different folds, and, in each iteration, we use one fold for testing, one for validation, and the remaining folds for training. In particular, we ensure that each patient is included once in the validation set and once in the test set by using the i-th iteration, the i-th fold as the test, and the previous one as the validation, as reported in Figure 4.

When training the CNNs, we augmented the dataset by applying random rotation and flipping. In particular, during the training, each volume underwent a vertical and horizontal flip and a rotation of an angle of either 90, 180, or 270 degrees with a probability of 0.5. The training set was balanced by replicating some randomly chosen volumes belonging to the minority class, which relies on data augmentation operations to introduce variability among the samples. Table 3 shows the details of the number of samples in the training set before and after the balancing phase and during the validation and test sets for each CV fold. Moreover, the extracted volumes were normalized in the (0,1) range to ensure that, in the classification step, the used convolutional neural networks operate with volumes that have the same scale across different patients. During the experiments, the maximum number of epochs was set to 500; the batch size was set to 16 for the SFB-NET and VB-NET and 32 for the 2DS-NET. The learning rate for the cross-entropy loss was set to 10^−6^. We used the Adam optimizer with a weight decay set to 10^−4^. To find the appropriate hyper-parameters, we implemented a grid search by varying the batch size in (8, 64), the learning rate in (10–7, 10–3), and the weight decay in (0, 10−4).

All experiments were carried out using Pytorch (version 1.10, developed by Meta AI and now part of the Linux Foundation umbrella), and the pre-processing step, including the different bounding box options, was implemented in MATLAB 2020b. 

## 4. Results

The clinical characteristics of the patients and the pathological and MRI features of each tumor are summarized in Table 4. The menopause status, grading, and class significantly differ (*p* < 0.05) between patients in cases of lymph node involvement. No other significant differences were observed between the two examined groups (LN+ vs. LN−).

Each of the performances of the different CNNs, based on each bounding option, is shown in Table 5. The best performances in terms of accuracy and AUC are obtained for the 2DS-NET: 78.63% and 77.86%, respectively. The 2DS-NET also showed the highest specificity; the highest sensibility was reported for the VB-NET based on the SVB and SIB as bounding options. Moreover, the solution involving the SIB option obtained the highest performance in terms of K.

Figure 5 shows the ROC curves of the implemented experiments, obtained by plotting the true positive rate against the false positive rate at various threshold settings. It is possible to note that when the SLVB and SLIB options are used, the model performs worse than when a random classifier is used.

Figure 6 shows the precision-recall curves of the implemented experiments that report the tradeoff between the precision and recall for different thresholds. When the classes are severely imbalanced, they are an effective indicator of how well the predictions worked. In particular, considering LN+ as the positive class and denoting with *tp*, *tn*, *fp*, and *fn* the true positive, true negative, the false positive, and the false negative, respectively, the precision and recall are computed as follows:(1)$$Precision=\frac{tp}{tp+fp}\;\;\;\;\;\;\;\;Recall=\frac{tp}{tp+fn}$$

In particular, the recall matches the sensitivity, while the precision represents the fraction of positive instances correctly classified among all the samples predicted as positive.

Finally, Figure 7 reports the confusion matrices computed by taking into account the predictions of the implemented models.

## 5. Discussion

It is now well established that the ALNS is directly dependent on tumor aggressiveness, which is reflected in a different representation of the breast lesion during DCE-MRI [12]. It has also been widely confirmed that the peritumoral tissue is influenced by tumor angio- and lympho-invasiveness [19,20,21]. Clinical and anamnestic data are actually not enough to determine the LNS with certainty, although some of these may benefit the doctors (in our case, cohort grading, class, and menopause status differ significantly between patients with lymph node involvement). In the majority of the works present thus far in the literature that exploits ML classifiers as predictors of the axillary cavity’s status in patients with breast cancer, generally only the features of the primary lesion have been evaluated, without considering the breast parenchyma adjacent to the lesions and, in particular, the tissue between the multiple lesions in cases of multifocality and multicentricity [13,15,16,26].

Furthermore, these works are highly heterogeneous in terms of the involved dataset sizes, feature extraction/selection methods employed, and trained classifiers. Most literature proposals extract the shape, texture, morphological, and first-order features, while pharmacokinetic parameters are exploited only by Liu et al. [27]. In our previous work [14,28] 3D extension of local binary patterns (LBPs) was also explored as features to enrich texture description. Regarding the proposed classifiers, they are predominantly support vector machines [29,30,31], logistic regressions [27,32,33], linear discriminant analyses [34,35], and random forests [14,36].

In our previous work [14,28], we combined patients’ clinical data, primary breast tumor histological information, and MRI radiomic features (First-Order, 3D Gray-Level Co-Occurrence Matrix, Three Orthogonal Planes-Local Binary Patterns) to predict the ALNS. In more detail, in a very innovative way, in this previous work, we considered both breast lesions and peritumoral regions for feature extraction considering a convex hull algorithm, which is the minimum volume bounded into a convex polygon and contains the ROI [14,28]. The high dimensionality of the problem to solve (257 features) makes the wrapper features selection method necessary before using a random forest (RF) classifier to provide the prediction.

In this work, we make a step forward by exploiting DL approaches and CNNs, in particular, for ALN status prediction. Their ability to autonomously learn the set of features that fits the specific task to solve makes the feature extraction step unnecessary, which is typical of ML techniques, thus overcoming the problem of finding the most discriminating set of primary tumor characteristics. Moreover, we also evaluate how the peritumoral region affects the performance of the involved networks by investigating several tumor bounding options that differ according to the amount of healthy tissue to be included.

The implemented methodology consists of three main steps: the *imaging data definition*, used to prepare data belonging to the DCE sequences of different patients; the volume extraction and bounding options, which describe the tumor bounding options; and the CNN architecture for the ALN assessment, which introduces CNNs that differ according to the characteristics of the volumes considered as input.

Among all of the experiments, the results obtained involving the SIB option and the VB-NET showed the best performance. We argue that the obtained result is completely in accordance with the specific problem to be solved. In the SFB option, the fixed-sized bounding box may result in the inclusion of an excessive amount of healthy tissue with respect to the lesioned one, especially in patients with a small tumor lesion. As a consequence, the SFB-NET may focus on areas of the input volumes that do not contain useful information for the prediction and do not reflect the tumor’s intrinsic behavior. The SVB and SIB options consider the smallest 3D cubical bounding box circumscribed to the tumor region, which limits the amount of the included peripheral and healthy tissue. As a result, the VB-NET directly takes the region of interest as input, leading to an increase in performance. However, the SVB extracts volumes whose voxels have different dimensions (in terms of mm) along the three spatial axes. In more detail, the spacing between slices is usually greater than the pixel spacing, resulting in a 3D cubical box whose dimension greatly depends on the x- or y-axes. As a consequence, the SVB option may introduce a high amount of healthy tissue along the z-axis, resulting in the need to introduce the SIB procedure that considers volumes with isotropic voxels. The considerations made between the SVB and SIB can be also used for the SLVB and SLIB. However, the results shown in Table 5 suggest that approaches that deal with a box for each lesion may not be the best solution. We argue that the presence of multiple lesions may be relevant information for the prediction of axillary lymph node metastasis. Therefore, if the lesions are split into different boxes, such information may be lost when considering, in particular, the multifocal and multicentric tumors as a single entity and not as different tumors. Finally, the 2DS option that creates a set of 3D slices for each patient, has the advantage of increasing the size of the dataset. However, when comparing the results with the solution involving the SIB procedure, it is possible to note a high difference in terms of sensitivity that makes us confirm the SIB as the preferred bounding option.

Moreover, Table 5 shows that the different bounding box options affect the results. In particular, the solutions involving the SIB and SVB options are the approaches with the more balanced results of the two classes, providing also the largest performances in terms of sensitivity (76.67% and 71.67%, respectively). On the contrary, the other bounding box options tend to favor the negative class (LN−); we deem that this cannot be considered a limitation of the work, but rather a result. Indeed, an explanation can be found in the following reasons:In the SFB option, the fixed-sized bounding box may result in an excessive amount of healthy tissue with respect to the lesioned one;We argue that the presence of multiple lesions may be relevant information for the prediction of axillary lymph node metastasis. Therefore, when the SLVB and SLIB options are used, such information may be lost;The 2DS option considers 2D slices and, thus, does not exploit volumetric features.

To the best of our knowledge, this paper represents the first work exploring the use of different primary tumor bounding options and convolutional neural networks to evaluate how the peritumoral region affects the ALNS assessment.

The work proposed by Nguyen et al. [25] represents the only one that has aimed to predict ALN metastasis using primary tumor DCE-MRI features with a DL approach. In that work, a 3D CNN is implemented to process the DCE-MRI images using a subtractive approach that works with the third, fourth, and fifth post-contrast volumes. A 3D cuboidal bounding box of the size 50 × 50 × 50, encompassing the tumor region, is used to crop the DCE-MRI data.

Nguyen et al. obtained the best values with 64.64%, 69.29%, 37.37%, and 58.56% for ACC, SPE, SENS, and AUC, respectively; all are lower compared to our best results (78.06%, 78.91%, 74.07%, and 75.84%), although they are not directly comparable since the datasets are different.

In our opinion, the fixed-size bounding box used in Nguyen et al. reduces the generalization capability of the implemented model. We also consider the absence of significant differences between our architectural model and that used by Nguyen et al. Furthermore, the authors included patients from two different hospitals, which may influence the imaging acquisition homogeneity.

Our work demonstrates that the peritumoral parenchyma, and, in particular, ones located between multiple lesions in the case of a multicentric or multifocal tumor, contains information not visible to the radiologist’s eye that correlates with the metastatic spread to the lymph nodes of the axillary cable. Parenchyma near the tumor lesions in current MRI clinical practice is not considered. In detail, in our work, the bounding box that has shown the best results is that which includes the parenchyma closest to the lesions and between multiple lesions but excludes the parenchyma furthest from the lesions themselves. These data suggest that there is an invisible cellular diffusion near the tumor lesions that artificial intelligence can help to reveal.

In our previous works, we have shown how the features contained in the primary tumor, first shown in [13,27], and in the peritumor edema, then explored in [19], significantly influence the LNS. The demonstration that the contribution of the peritumoral parenchyma is significant paves the way for new research, which has had little or no exploration in the current literature.

The main limitation of our study is the small number of patients that belong to a single medical center. Although the size of the dataset is similar to that used in other works [16,27,32,33], which analyzed samples in the range of 146–164 patients, a future perspective aim is to include data collected from different centers and also to evaluate how different image modalities contribute to the prediction of axillary lymph node metastasis. Another important limitation is the imbalance of class priors (22 patients with LN+ vs. 128 patients with LN−), which, in several cases, resulted in sensitivity values lower than the specificity ones, although we implemented standard techniques to cope with this issue.

In our future work, we want to implement data and test CNNs on different datasets and validate the bounding box technique on different kinds of images and MRI sequences. Furthermore, we want to investigate the role of the individual histological parameters reported in this article in relation to the radiomics data in the prediction of LNS.

However, these results are absolutely noteworthy in relation to the role played by the peritumoral parenchyma.

## 6. Conclusions

In conclusion, we demonstrated that a selective inclusion of peritumoral tissue increases the performance of the CNNs in predicting lymph node involvement and that the tissue localized among different tumoral lesions, especially in cases of multifocal and multicentric cancers, is strongly related to angio- and lymph-invasiveness and to lymph node metastasis. This tissue is barely evaluated by the radiologist’s eye on all MRI sequences and, in this way, an AI approach may represent a valid supporting tool.

Of course, more studies are needed to confirm these results. Future perspectives include increasing the number of patients, in particular patients with a positive LNS, to test the reproducibility of the results on different samples (different MRI and samples) and to include semantic features in the CNN analysis.

## Figures and Tables

**Figure 1 cancers-14-04574-f001:**
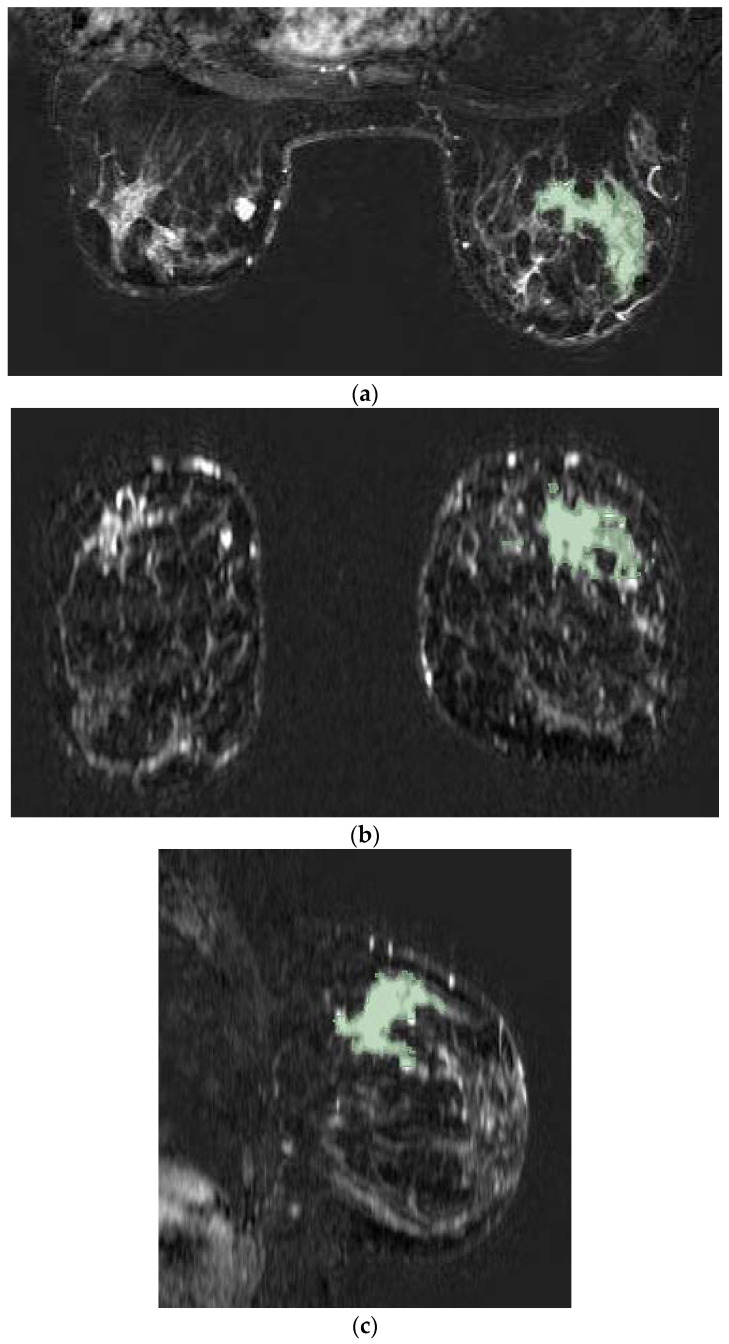
Tumor lesion segmentation using 3D Slicer software in axial (**a**), coronal (**b**), and sagittal (**c**) MRI projections during the second phase of the post-contrast sequence as demonstrated in a case involving a 56-year-old woman with right invasive ductal breast cancer with unifocal mass-like lesion characterized by spiculated margins and heterogeneous enhancement after contrast medium administration with curve SI/T type III.

**Figure 2 cancers-14-04574-f002:**
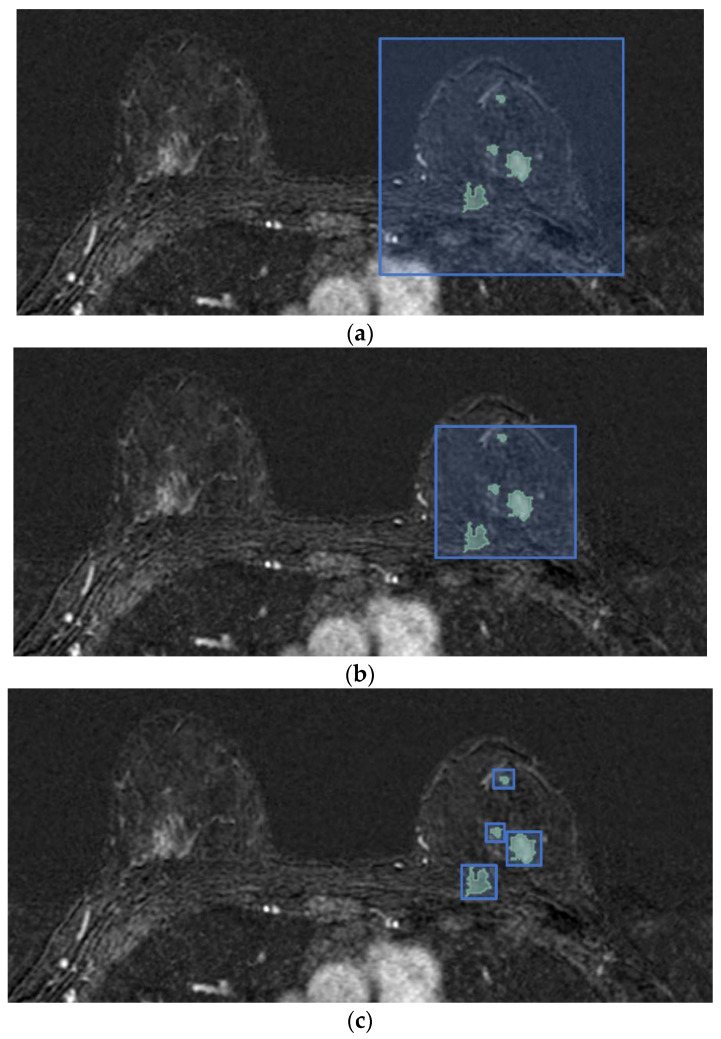
Differences in axial projections of the implemented tumor bounding options. (**a**) In the SFB, a fixed-size 3D bounding box is used; (**b**) in the SVB option, the smallest 3D bounding box circumscribed to the tumor region is considered, and (**c**) in the SLVB, the SVB option is applied to each lesion of the patient.

**Figure 3 cancers-14-04574-f003:**
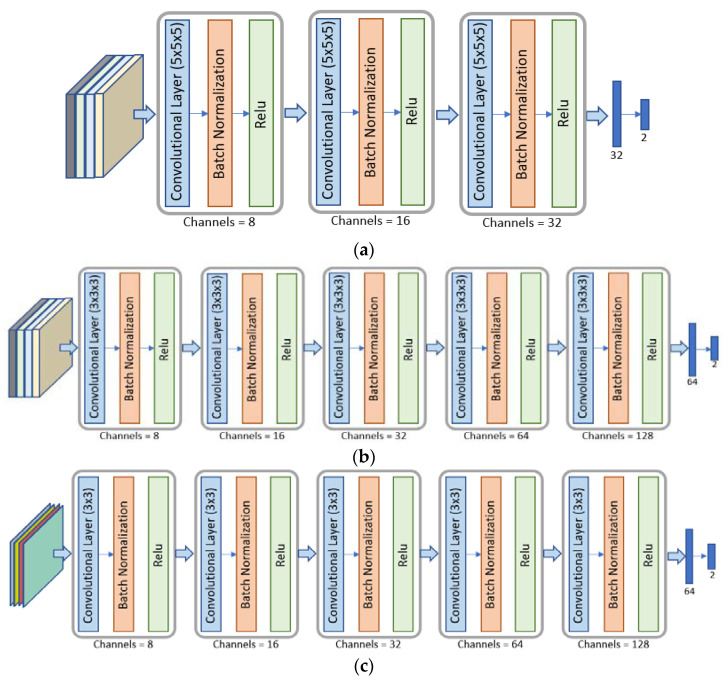
Architectures of the different CNNs used. Panel (**a**): The SFB-NET is a 3D CNN with three reduction blocks and two fully connected layers. Panel (**b**): The VB-NET is a 3D CNN with five reduction blocks and two fully connected layers. Panel (**c**): The 2DS-NET is a 2D CNN with five reduction blocks and two fully connected layers.

**Figure 4 cancers-14-04574-f004:**
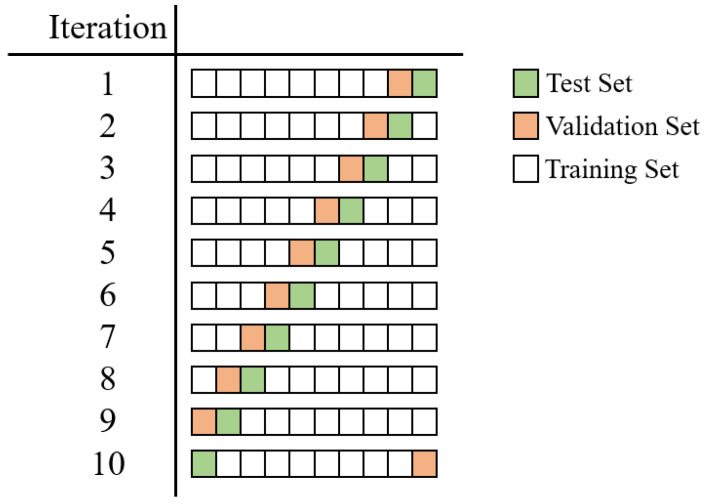
Details about the implemented patient-based 10-fold CV. In the i-th iteration, the i-th fold is selected as the test (green), and the previous one is selected as the validation (orange). The remaining folds are included in the training set.

**Figure 5 cancers-14-04574-f005:**
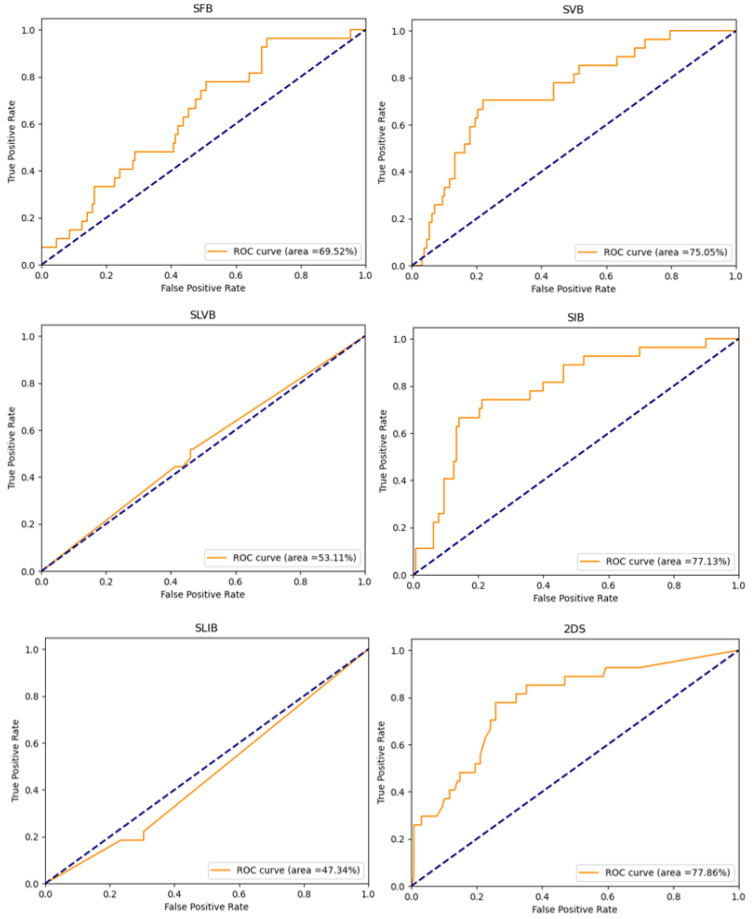
ROC curves of the implemented experiments. The title of each plot suggests the tumor bounding option that is used.

**Figure 6 cancers-14-04574-f006:**
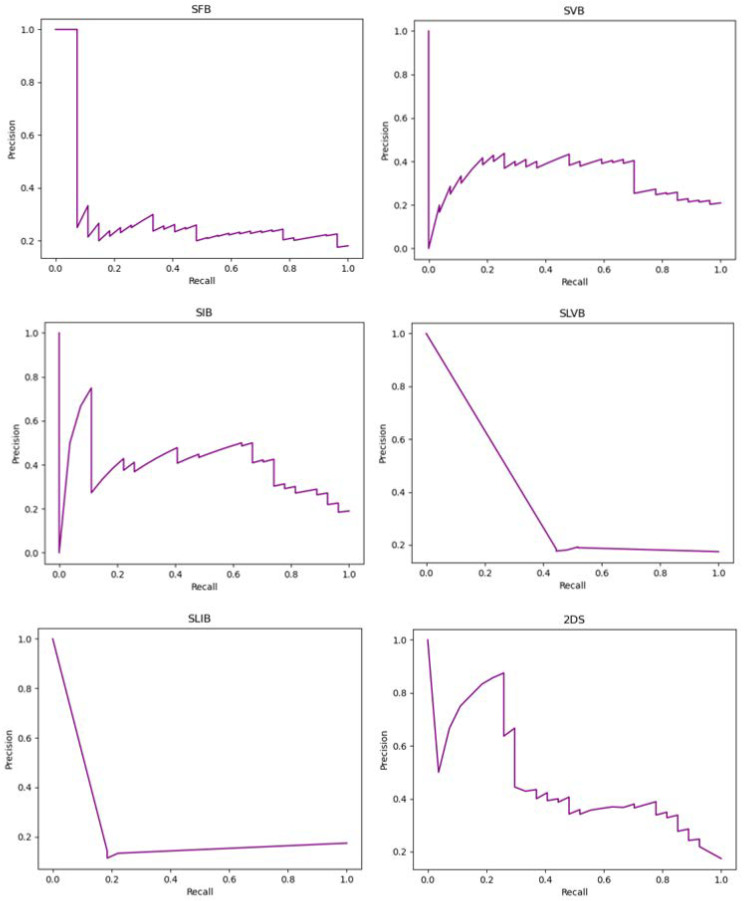
Precision-recall curves of the implemented experiments. The title of each plot suggests the tumor bounding option that is used.

**Figure 7 cancers-14-04574-f007:**
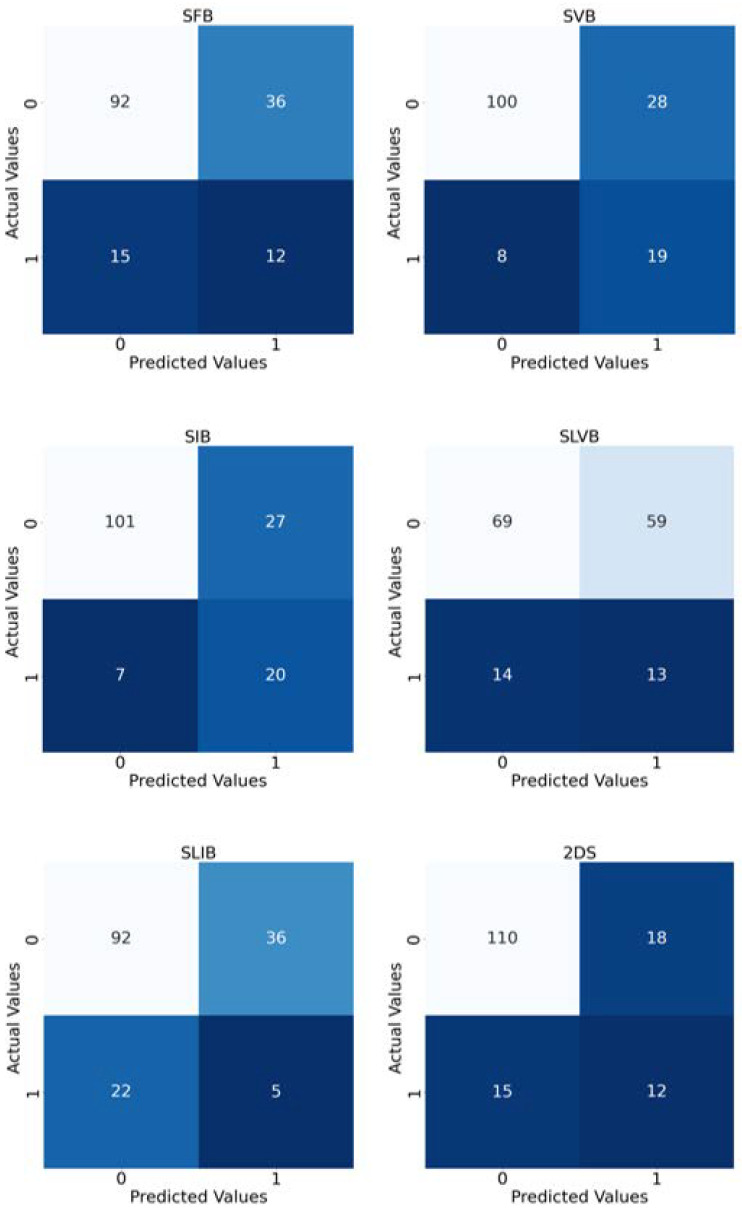
Confusion matrices of the implemented experiments. The title of each matrix suggests the tumor bounding option that is used. In order to have a matrix for each experiment, we merged the predictions of the 10 folds considered as test sets. The colour depends on the number inside the square: the higher the number, the lighter the colour.

**Table 1 cancers-14-04574-t001:** Details about the involved dataset.

Patients	153
BC lesion	155
LN+	128
LN−	27
Series	3D T1-weighted DCE
Mode	3T (Discovery 750; GE Healthcare, Milwaukee, WI, USA)
Dose	0.2 mmol/kg of Gadobenate-dimeglumine
Injection flow rate	2 mL/s

**Table 2 cancers-14-04574-t002:** Details of the proposed bounding box options.

Bounding Option	Details
SFB	A fixed-size 3D bounding box is used.
SVB	The smallest 3D cubical bounding box is used.
SIB	The SVB is applied after resizing the volumes to obtain MRI images with isotropic voxels.
SLVB	The SVB is applied to each lesion.
SLIB	The SLVB is applied after resizing the volumes to obtain MRI images with isotropic voxels.
2DS	The SVB is applied, and then the sequence of the 3D cropped volumes is cut along the projection with the highest spatial resolution.

**Table 3 cancers-14-04574-t003:** Details about the number of samples in the training, validation, and test sets.

Fold	Training Set		Balanced Training Set		Validation Set		Test Set	
	LN+	LN−	LN+	LN−	LN+	LN−	LN+	LN−
1	23	101	101	101	2	13	2	14
21	22	101	101	101	2	14	3	13
32	21	102	102	102	3	13	3	13
43	21	102	102	102	3	13	3	13
54	21	102	102	102	3	13	3	13
65	21	103	103	103	3	13	3	12
76	21	104	104	104	3	12	3	12
87	21	104	104	104	3	12	3	12
98	22	103	103	103	3	12	2	13
109	23	102	102	102	2	13	2	13

**Table 4 cancers-14-04574-t004:** Patient and tumor feature frequencies and relative percentages are reported in relation to the final label (lymph node LN− involvement). The difference between the two groups (LN+ vs. LN−) was reported, and the statistical significance was set at 0.05 (*). HT (hormonotherapy), IS curve/T (intensity signal curve/time), IDC (invasive ductal carcinoma), ILC (invasive lobular carcinoma), TN (triple negative).

Class	Group	Frequency	Percentage	LN+	LN−	*p* Value
Familiarity	none	109	70.32%	22	87	0.1314
	≥1 fam	46	29.68%	5	41	
HT	no	141	90.97%	27	114	0.0733
	yes	14	9.03%	0	14	
Menopause	no	67	43.23%	17	50	0.0233 *
	yes	88	56.77%	10	78	
IS curve/T	I	21	13.55%	3	18	0.2819
	II	69	44.52%	10	59	
	III	65	41.94%	14	51	
Margins	regular	7	4.52%	0	7	0.5504
	irregular	83	53.55%	18	65	
	lobulated	19	12.26%	3	16	
	spiculated	46	29.68%	6	40	
Histotype	IDC	129	83.23%	23	106	0.7351
	ILC	23	14.84%	4	19	
	Medullary	3	1.94%	0	3	
Grading	1	21	13.55%	1	20	0.0011 *
	2	69	44.52%	7	62	
	3	65	41.94%	19	46	
Class	Luminal A	61	39.35%	6	55	0.0013 *
	Luminal B	67	43.23%	9	58	
	Her2	12	7.74%	6	6	
	TN	15	9.68%	6	9	

**Table 5 cancers-14-04574-t005:** Performance of the CNNs in LNS prediction (LN+ vs. LN−). ACC (accuracy), SPE (specificity), SENS (sensibility), AUC (area under the curve), K (Cohen’s kappa coefficient), SFB (single fixed-size box), SVB (single variable-size box), SIB (single isotropic-size box), SLVB (single lesion variable-size box), SLIB (single lesion isotropic-size box), 2DS (two-dimensional slice), and NET (network). The best test performances are evident in bold.

Model	Option	ACC	SPE	SENS	AUC	K
		Mean	Mean	Mean	Mean	Mean
SFB-NET	SFB	70.62%	75.92%	43.33%	69.52%	0.1349
VB-NET	SVB	76.79%	78.17%	**71.67%**	75.05%	0.3753
VB-NET	SIB	78.13%	78.82%	**76.67%**	77.13%	**0.4487**
VB-NET	SLVB	52.71%	53.24%	48.33%	53.11%	0.0124
VB-NET	SLIB	62.58%	71.79%	18.33%	47.34%	−0.0798
2DS-NET	2DS	**78.63%**	**85.75%**	46.67%	**77.86%**	0.2911

## Data Availability

The data and the code used for the experiments are available on request.

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
