# Peer review of "CNN-Based Approaches with Different Tumor Bounding Options for Lymph Node Status Prediction in Breast DCE-MRI"

_cancers, 2022, doi:10.3390/cancers14194574_

Round 1
Reviewer 1 Report
This research work is devoted to the exploitation of three CNN models with Different Tumor Bounding Options in order to predict Lymph Node Status in Breast DCE-MRI. In particular, due to the fact that peritumoral tissue contains valuable information on tumor aggressiveness, six different tumor bounding options to predict the LNS using a CNN were analyzed. The work is overall well-organized and written. However, there are some concerns and parts that can be improved after a major revision.
In introduction, the state of the art and the challenges of Lymph Node prediction in breast cancer are not clearly reported. The innovation of this research work is missing.
In section 2, authors need to assign a clear methodological framework with steps to clarify the proposed DL methodology. The steps of data processing, training, validation and testing are missing. There is no information on the type of data used as input to the CNN model. Are the data available to the reader or the data and the codes available to a github? This could be really useful for the purposes of this study.
How the architectures of the CNN models have been selected? Information is missing. What about the resize options?
The results seems to be promising for the future work. However, in the discussion section, the authors should further clarify the usefulness of these results, their clinical impact (it is not defined).
The authors claim that “More in detail, in a very innovative way, we considered both breast lesions and peritumoral regions for feature extraction considering a convex hull algorithm.” There is no previous reference to this algorithm and how the innovation is emerged.
In discussion, it is claimed that there are no relevant previous works, except Nguyen et al. [30] to address the same problem. A more thoroughly literature exploitation is needed to show the innovation and impact of this work in prediction of lymph nodes in breast DCE-MRI.
The limitations are clear; however it is not fully defined how these limitations will be addressed in the future work. No clinical impacts are reported.
Author Response
Reviewer 1 reply
This research work is devoted to the exploitation of three CNN models with Different Tumor Bounding Options in order to predict Lymph Node Status in Breast DCE-MRI. In particular, due to the fact that peritumoral tissue contains valuable information on tumor aggressiveness, six different tumor bounding options to predict the LNS using a CNN were analyzed. The work is overall well-organized and written. However, there are some concerns and parts that can be improved after a major revision.
In introduction, the state of the art and the challenges of Lymph Node prediction in breast cancer are not clearly reported. The innovation of this research work is missing.
Thanks to the reviewer, two paragraphs have been added following these suggestions.
In section 2, authors need to assign a clear methodological framework with steps to clarify the proposed DL methodology. The steps of data processing, training, validation and testing are missing. There is no information on the type of data used as input to the CNN model. Are the data available to the reader or the data and the codes available to a github? This could be really useful for the purposes of this study.
Thanks to the reviewer for this comment that allows us to better explain our methodology. We summarized in Section 3 the proposed methodology, improving section 2 explaining the involved dataset. We inserted in section 3.4 more details about the training, validation and test steps. We reported in the "data availability" section (after the discussion and before the references) that the data and the code used for the experiments will be available on request.
How the architectures of the CNN models have been selected? Information is missing. What about the resize options?
Thansk to the reviewer for this comment. We included this information in section 3.4
The results seem to be promising for future work. However, in the discussion section, the authors should further clarify the usefulness of these results, their clinical impact (it is not defined).
Thanks to the reviewer for this comment which allows us to be more meaningful. The following paragraphs have been added in the discussion: “Our work demonstrates that the peritumoral parenchyma, and in particular the one located between multiple lesions in the case of multicentric or multifocal tumor, contains information not visible to the radiologist's eye which correlate with the metastatic spread to the lymph nodes of the axillary cable. The parenchyma near the tumor lesions in MRI current clinical practice is not considered. In detail, in our work, the bounding box that has shown the best results is that which includes the parenchyma closest to the lesions and between multiple lesions but excludes the parenchyma furthest from the lesions themselves. This data suggests that there is an invisible cellular diffusion near the tumor lesions but that artificial intelligence can help to reveal.
In our previous works we have shown how the features contained in the primary tumor, first [8, 20], and in the peritumor edema, then [14], influence significantly the LNS. The demonstration that the contribution of the peritumoral parenchyma is significant opens the way for new research which has so far been little or no exploration in the current literature”.
The authors claim that “More in detail, in a very innovative way, we considered both breast lesions and peritumoral regions for feature extraction considering a convex hull algorithm.” There is no previous reference to this algorithm and how the innovation is emerged.
To be clearer we specified that the sentence was referring to our previous work and we added the references. Anyway the convex hull was explained in a new sentence.
In discussion, it is claimed that there are no relevant previous works, except Nguyen et al. [30] to address the same problem. A more thoroughly literature exploitation is needed to show the innovation and impact of this work in prediction of lymph nodes in breast DCE-MRI.
Actually there are very few works about this topic and we reported this lack in our previous work entitled: “Radiomics MRI for lymph node status prediction in breast cancer patients: the state of art”. However, following reviewer’ suggestion we reported the following sentence in the discussion: “As reported in our recent review about lymph node status in breast cancer patients, MRI and Artificial Intelligence, there are very few works about this topic [13]. In this review the main role of DCE-MRI is demonstrated compared to the other sequences. In particular, the work proposed by Nguyen et al. [36] represents the….”
The limitations are clear; however it is not fully defined how these limitations will be addressed in the future work. No clinical impacts are reported.
Thanks to the reviewer for this suggestion. We added some specific sentences in the discussion.

Reviewer 2 Report
The authors used a CNN-based deep learning approach to predict Lymph Node Status Prediction in Breast DCE-MRI with Different Tumor Bounding Options. However, the relationship between LN+ and LN- with DCE-MRI images is not clear. Moreover, the importance of different bounding boxes should be explained explicitly. This study needs technical and structural improvements. Some of my suggestions are provided below:
1. Rewrite the title in a more meaningful way and without colons.
2. It is better to be specific rather than using the terms like convolutional neural networks, deep learning, and artificial intelligence altogether, for instance, in the title as well as the simple summary.
3. The introduction section needs improvement. In the last paragraph, please mention categorically the contribution and motivation behind this study.
4. Explain the inclusion and exclusion criteria used. Also mentioned how many radiologists and/or pathologists were involved.
5. In subsection 2.3, the authors mentioned 153 patients (with 155 malignant lesions), whereas, in subsection 2.5, the dataset accounts for 27 positive (LN+) and 128 negatives (LN-) patients. How? LN+ and LN- represent patients or MRI images?
6. How LNS were dichotomized into LN+ AND LN-?
7. Make a Table to summarize the dataset.
8. All the six bonding options should be summarized in Table.
9. Provide most relevant references in subsections 2.2: MRI examination, 2.3: clinical data, 2.4: histological data, 2.5: axillary lymph node status, 2.6: preprocessing, and 2.7: segmentation.
10. Subsection 2.8: Deep learning Analysis and its subsubsections 2.8.1, 2.8.2, and 2.8.3 should be a separate section. Moreover, explanations given about box options in section 3: experimental setup should be adjusted in this new section.
11. It is crucial to focus on the preparation of the dataset. It is unclear how the bounding options were selected.
12. It is unclear how the authors selected optimum hyperparameters. Also, details about data augmentation are missing.
13. How dataset was balanced before training. Please make a table of original and augmented images used for training CNN models.
14. What are the input dimensions for every CNN model? Also, mention the outputs of every CNN model.
15. The importance of Table 1 and results section should be explained thoroughly.
16. Provide ROC-AUC and PR curves for all the models, and compare them.
17. Provide accuracy and loss curves, together with confusion matrices.
18. As the dataset is imbalanced, it is advised to calculate Cohen’s kappa score.
19. It is advised to review the writing and spelling typos throughout the manuscript.
20. What does Table 2 represent? The classification of DCE-MRI or LN+ and LN-?
Author Response
Reviewer 2 reply
The authors used a CNN-based deep learning approach to predict Lymph Node Status Prediction in Breast DCE-MRI with Different Tumor Bounding Options. However, the relationship between LN+ and LN- with DCE-MRI images is not clear. Moreover, the importance of different bounding boxes should be explained explicitly. This study needs technical and structural improvements. Some of my suggestions are provided below:
- Rewrite the title in a more meaningful way and without colons.
Thanks for this suggestion. The title has been modified as following “Convolutional Neural Network-Deep Learning Approaches with Different Tumor Bounding Options for Lymph Node Status Prediction in Breast DCE-MRI”
- It is better to be specific rather than using the terms like convolutional neural networks, deep learning, and artificial intelligence altogether, for instance, in the title as well as the simple summary.
Thank you for your suggestion. We modified the title and the summary.
- The introduction section needs improvement. In the last paragraph, please mention categorically the contribution and motivation behind this study.
Introduction section has been improved providing the required information.
- Explain the inclusion and exclusion criteria used. Also mentioned how many radiologists and/or pathologists were involved.
Inclusion and exclusion criteria were added in the M&M section as suggested by the reviewer. the following sentences were reported in the 2.1 and 2.3 sections, respectively: “The images were analyzed by two radiologists with 10 and 3 years of experience respectively. ” and “an anatomic-pathologist with more than 15 years of experience”
- In subsection 2.3, the authors mentioned 153 patients (with 155 malignant lesions), whereas, in subsection 2.5, the dataset accounts for 27 positive (LN+) and 128 negatives (LN-) patients. How? LN+ and LN- represent patients or MRI images?
Thanks to the reviewer for this comment which allows us to be clearer. The following sentence has been added in M&M section: “A total of 153 patients (average age 55 years; range 30-85) met the inclusion criteria. In two patients who had bilateral breast cancer the two breasts have can been considered as a single independent part. Then a total of 155 malignant breast cancer lesions were included totally.
- How LNS were dichotomized into LN+ AND LN-?
Thanks to the reviewer for this comment which allows us to be clearer. The following sentence has been added in M&M section: “The LNS status was assessed as positive if there was at least one lymph node involved by metastasis at definitive histopathological axillary cable sample (LN+); LNS was considered as negative if all axillary lymph nodes were safe (LN-).”
- Make a Table to summarize the dataset.
We inserted Table 1 that summarizes the dataset.
- All the six bonding options should be summarized in Table.
Thanks for the suggestion. We added Table 2 that summarizes the proposed bounding box options
- Provide most relevant references in subsections 2.2: MRI examination, 2.3: clinical data, 2.4: histological data, 2.5: axillary lymph node status, 2.6: preprocessing, and 2.7: segmentation.
The relative references have been added to the subsections 2.4, 2.5, 2.6, 2.7, following reviewer’ suggestion. However there are no specific references for subsections 2.2 and 2.3 being original from our center.
- Subsection 2.8: Deep learning Analysis and its subsubsections 2.8.1, 2.8.2, and 2.8.3 should be a separate section. Moreover, explanations given about box options in section 3: experimental setup should be adjusted in this new section.
Thanks for this suggestion. We organized Deep Learning Analysis, Imaging data definition, Volumes extraction and Bounding Options, and CNNs architecture in Section 3, Subsection 3.1, 3.2 and 3.3 respectively. Then we included the Experimental Setup in subsection 3.4
- It is crucial to focus on the preparation of the dataset. It is unclear how the bounding options were selected.
We reported the required information in section 3.2
- It is unclear how the authors selected optimum hyperparameters. Also, details about data augmentation are missing.
Thanks for this suggestion. We included these details in the experimental setup.
- How dataset was balanced before training. Please make a table of original and augmented images used for training CNN models.
Thanks for this suggestion. We included in Table 3 the details about the number of samples in the train, validation and test sets.
- What are the input dimensions for every CNN model? Also, mention the outputs of every CNN model.
We clarify the requested details in section 3.3 and 3.4.
- The importance of Table 1 and results section should be explained thoroughly.
Thanks to the Reviewer for this comment. The histological parameters and their role in the LNS prediction, have been just cited in the first part of the discussion. However we introduced a more specific sentence in the future perspectives paragraph. This point, in fact, is a very relevant topic that we are already analyzing for a subsequent work.
- Provide ROC-AUC and PR curves for all the models, and compare them.
Thanks to the reviewer for this comment that let us improve the manuscript. We included the ROC and PR curves in figures 5 and 6 respectively.
- Provide accuracy and loss curves, together with confusion matrices.
We report the confusion matrices in figure 7. However, in our opinion, the inclusion of accuracy and loss curves may introduce a lot of images, making the paper difficult to read. Moreover, the training curves do not suggest results worth reporting in the paper. Since all the experiments run in 10 fold-CV, we have 10 different loss and accuracy curves for each model and bounding option, resulting in 20 images for each experiment. We provide the accuracy and loss curves in the review response. We argue that the small size of the validation set (15 or 16 samples) causes the loss and accuracy curves to fluctuate. Indeed, the variation in the prediction of a few samples has a strong impact on both accuracy and loss. As reported in Section 3.4, the maximum number of epochs is set to 500. However, according to the training curves, it is possible to note that the optimal epoch is represented by a significantly smaller number.
- As the dataset is imbalanced, it is advised to calculate Cohen’s kappa score.
Thanks for the suggestion. We added the Cohen’s Kappa score for each experiment
- It is advised to review the writing and spelling typos throughout the manuscript.
The manuscript has been reviewed following this suggestion.
- What does Table 2 represent? The classification of DCE-MRI or LN+ and LN-?
Thanks for the suggestion. The title of the table was modified to make the classification clearer

Round 2
Reviewer 2 Report
Dear Authors:
I appreciate your efforts in the revised manuscript. However, it can be concluded that the current results cannot be accepted for publication. All the "six bounding options" worked well merely for class. So, I recommend you improve the results and resubmit it when it worked reasonably well for both classes.
Regards
Author Response
REPLY
Dear reviewer,
We thank you for all the suggestions you provided in the first round of the revision that helped us improve the quality of the manuscript.
About the second round we wish to clarify that, as we reported in the paper, inspired by few papers in the literature that suggested the importance of the peritumoral parenchyma [1,2,3], we move forward with respect to our previous work [4,5] offering the technical contributions now listed at the end of section 1. Furthermore, the results of this manuscript are original both for the study design and, above all, for the proposal of a deep learning-based approach to the prediction of axillary lymph node metastasis. Indeed, there is a lack of deep learning-based approaches in the current literature.
Our work aims to conduct a study on the influence of non-tumour tissue, rather than to propose a solution to the problem of predicting lymph node metastasis. In the literature, this work is the first attempt to explore different bounding box options to evaluate the influence of the inclusion of features on a specific output.
Table 5 shows that the different bounding box option affects the results. In particular, the solutions involving the SIB and SVB options are the approaches with more balanced results among the two classes, providing also the the largest performance in terms of sensitivity (76.67% and 71.67%, respectively), differently from what you state in the sentence “All the six bounding options worked well merely for class”. On the contrary the other bounding box options tend to favor the negative class (LN-, absence of metastasis): we deem that this cannot be considered a limitation of the work, but rather a result. Indeed, we argue that the obtained results agree with the specific problem to solve for the following reasons :
- In the SFB option, the fixed-sized bounding box may result in an excessive amount of healthy tissue with respect to the lesioned one, especially in patients with a small tumour region.
- The SVB and SIB options consider the smallest 3D cubical bounding box circumscribed to the patient’s tumor region, limiting the amount of healthy tissue to include. As a result, the involved CNN directly takes the region of interest as input, leading to an increase in performance. However, the SVB extracts volumes whose voxels have different dimensions (in terms of mm) along the three spatial axes. As a consequence, it may introduce a high amount of healthy tissue along the z-axis, resulting in the need of introducing the SIB procedure that considers volumes with isotropic voxels.
- When the SLVB and SLIB options are used, we argue that the presence of multiple lesions may be relevant information for the prediction of axillary lymph node metastasis. Therefore, if the lesions are split into different boxes, such information may be lost.
- The 2DS option considers 2D slices, thus not exploiting volumetric features.
(we modified the discussion of the manuscript and added a bulleted list to make easier the reading)
The strong degree of balance in the dataset increases the difficulty of the problem under consideration making the obtained performance really good and encouraging.
Although it is not possible to perform a comparative analysis with the literature, since our paper represents a unique work of its kind, it is possible to note that the results are close to those obtained so far in the predictive analysis of the lymph node status in breast cancer patients with other radiomic approaches [6][7].
[1]Santucci, D., Faiella, E., Cordelli, E., Calabrese, A., Landi, R., de Felice, C., Beomonte Zobel, B., Grasso, R.F., Iannello, G., Soda, P.: The impact of tumor edema on t2-weighted 3t-mri invasive breast cancer histological characterization: A pilot radiomics study. Cancers 13(18), 4635 (2021)
[2]Sun, Q., Lin, X., Zhao, Y., Li, L., Yan, K., Liang, D., Sun, D., Li, Z.C.: Deep learning vs. radiomics for predicting axillary lymph node metastasis of breast cancer using ultrasound images: don’t forget the peritumoral region. Frontiers in oncology 10, 53 (2020)
[3]Krizhevsky, A., Sutskever, I., Hinton, G.E.: ImageNet classification with deep convolutional neural networks. In: Pereira, F., Burges, C.J.C., Bottou, L., Weinberger, K.Q. (eds.) Advances in Neural Information Processing Systems 25, pp. 1097–1105. Curran Associates, Inc. (2012)
[4] Cordelli, E., Sicilia, R., Santucci, D., de Felice, C., Quattrocchi, C.C., Zobel, B.B., Iannello, G., Soda, P.: Radiomics-based non-invasive lymph node metastases prediction in breast cancer. In: 2020 IEEE 33rd International Symposium on Computer-Based Medical Systems (CBMS). pp. 486–491. IEEE (2020)
[5] Santucci, D., Faiella, E., Cordelli, E., Calabrese, A., Landi, R., de Felice, C., Beomonte Zobel, B., Grasso, R.F., Iannello, G., Soda, P.: The impact of tumor edema on t2-weighted 3t-mri invasive breast cancer histological characterization: A pilot radiomics study. Cancers 13(18), 4635 (2021)
[6] Nguyen, S., Polat, D., Karbasi, P., Moser, D., Wang, L., Hulsey, K., C ̧obanog ̆lu, M.C., Dogan, B., Montillo, A.: Preoperative prediction of lymph node metastasis from clinical DCE MRI of the primary breast tumor using a 4D CNN. In: International Conference on Medical Image Computing and Computer-Assisted Intervention. pp. 326–334. Springer (2020)
[7]Calabrese, A., Santucci, D., Landi, R., Zobel, B.B., Faiella, E., de Felice, C.: Radiomics MRI for lymph node status prediction in breast cancer patients: the state of art. Journal of Cancer Research and Clinical Oncology pp. 1–11 (2021)

Round 3
Reviewer 2 Report
Dear Authors:
Thank you for updating the manuscript. It could be better if you release the code and dataset so that the research community could further work on this approach.
Regards
Author Response
Thanks to the Reviewer for all the suggestions.
Certainly. Both the code and the data will be available upon request. We will specify this in the dedicated paragraph "data availability"
